# Factors influencing the adoption of the BYOD policy in teaching hospitals: A cross-sectional study from Southeastern Iran

Jahanpour Alipour[1], Abolfazl Payandeh[2], Afsaneh Karimi[3]*

1 Health Human Resources Research Center, School of Health Management and Information Sciences, Shiraz University of Medical Sciences, Shiraz, Iran, 2 Genetics of Non-Communicable Disease Research Center, Zahedan University of Medical Sciences, Zahedan, Iran, 3 Pregnancy Health Research Center, Zahedan University of Medical Sciences, Zahedan, Iran

* afsanehkarimi2014@gmail.com

## Abstract

### Introduction

Clinicians are increasingly using their devices for work at hospitals, a practice known as Bring-your-own-device (BYOD), to enhance productivity and mobility. This study aimed to determine the affecting factors of intention to adoption of BYOD policy in public hospitals from the healthcare staff's perspective.

### Methods

A cross-sectional analytical study was done in 2024. The study population comprised 1130 healthcare workers from five teaching hospitals. A researcher-made and validated questionnaire was distributed among 620 samples. Data were analyzed by SPSS software using descriptive (mean and standard deviation) and analytical (Pearson and Spearman correlation test) statistics.

### Results

The mean score of facilitating conditions, perceived cost-effectiveness, perceived trust, perceived usefulness, perceived ease of use, and intention to adoption BYOD was $3.90 \pm 0.87$, $3.87 \pm 0.97$, $3.83 \pm 0.93$, $3.76 \pm 1.01$, $3.07 \pm 0.48$ and $3.62 \pm 1.16$, respectively. There was a positive significant correlation between factors of perceived usefulness, perceived ease of use, perceived cost effectiveness, perceived trust, and facilitating conditions with an intention to adoption the BYOD policy ($P < 0.05$).

### Conclusion

Healthcare workers have partially intended to adopt the BYOD policy. Ensuring the security of access to healthcare information, provision, support and maintenance of

**Data availability statement:** All relevant data are within the paper and its Supporting Information files.

**Funding:** The author(s) received no specific funding for this work.

**Competing interests:** The authors declare that they have no competing interests.

devices used by staff in the workplace for job-related activities can play a significant role in promoting the intention to adoption the BYOD. The results of the present study can be useful for planning and policy-making to increase the adoption and acceptance of the BYOD method in hospitals.

## Introduction

Using health information technology in personal health management and healthcare delivery improves health service quality, reduces medical errors, containments healthcare costs, and increases the efficiency and productivity of healthcare services [1–3]. About five percent of healthcare institutions' budgets are spent on information technology. However, deploying new technologies is not always beneficial and may not be responsive to costs. Failure to use the technologies can have negative consequences for healthcare organizations, patients, and staff [4–6]. The growth of information and communication technologies around the world has accelerated the process of using personal devices for doing organizational work. Using this technology by changing the organization's operations enhances the effectiveness of staff activities in the healthcare organization [7,8].

The use of mobile phones in Iran has had a positive role in the medication adherence of patients and in facilitating the provision of caregivers' care delivery [9,10]. Using bring your own devices (BYOD) is a whole new concept in the health industry that empowers employees to use personal devices (including mobile phones, laptops, personal digital assistants, and notebooks) to access organizational information resources and perform job-related tasks. Applying BYOD can impose the cost of purchasing, maintaining, and updating the device for the end user, thereby reducing organization costs [11–14]. In hospitals, healthcare professionals can utilize their own devices for work, such as accessing patient records and completing job-related tasks, through the BYOD policy [10,15,16]. In Australia, Wani et al. [16] revealed that healthcare staff use BYOD in their various tasks and prefer smartphones over other devices. The results of previous studies indicate that the BYOD policy has been successful and its adoption has reduced organizational infrastructure costs and increased employee satisfaction, mobility, and productivity in the institutions [7,10,11,17–24].

Mobile health technologies are used to monitor chronic diseases, weight control, blood pressure, and blood glucose [25]. Previous studies indicate that mobile technology has also been used in Iran mainly to manage diabetes, cancer, cardiovascular disease, patient education, weight control, asthma, depression, dialysis, gynecology, and iodine therapy respectively [9]. Increased mobility, flexibility, productivity, staff satisfaction, cost containment, accessibility and portability of used devices, ability to follow patient care in real-time, and decreased hospital admission are advantages of BYOD adoption in healthcare organizations [11,19,20,26,27]. Generally, healthcare professionals in Iran, including physicians and nurses, have a positive attitude toward the use of mobile phones in providing healthcare services [28,29].

However, previous studies indicate that the use of emerging technologies in the field of healthcare poses challenges. Therefore, consideration and study of the factors affecting the adoption of such technologies are essential to prevent or reduce the unintended negative consequences for patients, providers, and organizations [30]. The technology acceptance model (TAM) provides a valid and robust framework for exploring the acceptance or rejection of a new technology [31–33]. At the various versions of TAM factors such as perceived usefulness, perceived ease of use, perceived trust [14], perceived cost-effectiveness, intention to accept [26,34] and facilitating conditions [35] have been identified as factors influencing users' acceptance of the BYOD policy. Therefore, this study aimed to explore the influencing factors on the intention to adoption of BYOD policy in five teaching hospitals of Zahedan University of Medical Sciences.

## Methods

### Study design and setting

The descriptive – analytical, cross-sectional study was conducted in 2024 at all five teaching hospitals affiliated with Zahedan University of Medical Sciences, including Ali-Ibne-Abitaleb (a general hospital), Khatam-Ol-Anbia (a trauma center), Alzahra (a specialized ophthalmology hospital), Baharan (a specialized psychiatric hospital), and Buali (a specialized infections disease hospital). This study was approved by the Ethics Committee of the Deputy of Research and Technology of Zahedan University of Medical Sciences (No: IR.ZAUMS.REC.1399.452). Prior to filling out the questionnaire, all participants gave their consent both orally and in writing.

### Study population, sample size, and sampling method

The study population consists of 763 nurses, 217 physicians, 32 laboratory science, 35 radiology, 19 pharmacy, and 62 medical record/ health information technology workers. Only, for the nurse staff, 253 individuals were selected as the sample based on the Cochran formula. Regarding the rest of the employee groups, all employees of each group were considered due to the limitation of their number. Thus, a sample of 409 eligible individuals was selected using convenience (for nurses) and census (for physicians, laboratory science, radiology, pharmacy, and medical record/ health information technology workers) sampling methods. The data were collected using face-to-face interviews with an educated, experienced health technician familiar with the local language and accent.

### Data collection tool

A researcher-made questionnaire based on previous studies on factors affecting the adoption of BYOD policy based on various versions of TAM applied in the health domain [19,24,26,34,35] was used for data collection. The questionnaire consisted of two main sections: demographic information and factors affecting the intention to adoption of BYOD method. The first part of the tool contained five questions, including gender, age, work experience, job, and education. The second part of the questionnaire comprised 32 questions in six dimensions of perceived usefulness (n = 8), perceived ease of use (n = 10), perceived trust in using BYOD (n = 4), facilitating conditions (n = 5), perceived cost-effectiveness (n = 3) and intention to adoption of BYOD (n = 2). Each question was scored by the respondents for importance. For each question, a five-point Likert scale (from 1: very low to 5: very high) was used to rate each sub-factor.

To assess the reliability of the tool, the questionnaire was administered to 20 subjects in the 10-day interval and confirmed by Cronbach's alpha correlation test (0.82). The validity of the questionnaire was confirmed by the opinions of eight subject experts, including six experts in Health Information Management and two experts in Health informatics. All of them had PhD degrees. Since the number of experts was 8, if the content validity ration (CVR) score was higher than 0.78 and the content validity index (CVI) score higher than 0.70, the validity of the questions was confirmed [36]. Therefore, the tool was confirmed by CVR and CVI scores of 0.92 and 0.88, respectively.

## Data analysis

Data were analyzed using the SPSS software version 22 by applying descriptive (mean, standard deviation, and percent) and analytic (Pearson and Spearman correlation test) statistics. A minimum of 1 and a maximum of 5 were possible for a mean of each factor. Hence, to assess the staff's perspective regarding the factors affecting the intention to adoption of the BYOD method, a mean score of less than 2.5 indicates weak influence factors, a score of 2.5 to less than 3.75 denotes moderate influential factors, and a score of 3.75 to 5 indicates strong influence. The correlations between research variables were analyzed using Pearson correlation (r). A value of r = below 0.39, 0.40–0.69, and 0.70–1.00 indicated weak, moderate, and strong correlations, respectively [37]. The P-value <0.05 was considered significant between variables.

## Results

Of the 620 distributed questionnaires, 410 were completed. Ten returning questionnaires related to the nursing group were excluded from the study due to defects in more than 20% of the items needed to be completed. Therefore, data analysis was performed based on the remaining 400 complete questionnaires.

Nearly two-thirds of the respondents were female (61%). Most of them were between 23–30 years old and had 2–7 years' experience with 32.8% and 33%, respectively. Little more than half of the respondents were nurses (53.8%) and about two-thirds of respondents had a bachelor's degree (63%). (Table 1)

Facilitating conditions and perceived cost-effectiveness of the BYOD policy were identified as the most influential factors for adopting the BYOD policy. Staff had a relatively high intention to adopt the BYOD policy at hospitals. Perceived

**Table 1. Demographic characteristics (n = 400).**

| Category | Subcategory | Frequency | |
| --- | --- | --- | --- |
| | | **Number** | **Percent** |
| **Gender** | Female | 244 | 61 |
| | Male | 165 | 39 |
| **Age (year)** | 23-30 | 131 | 32.8 |
| | 31-37 | 99 | 24.8 |
| | 38-45 | 115 | 28.8 |
| | 46-52 | 55 | 13.8 |
| **Work experience (years)** | ≤ 1 | 25 | 6.3 |
| | 2-7 | 132 | 33 |
| | 8-14 | 111 | 27.8 |
| | 15-21 | 89 | 22.3 |
| | 22-28 | 43 | 10.8 |
| **Job** | Physician | 60 | 15 |
| | Nursing | 215 | 53.8 |
| | Medical Record/ HIT (Health Information Technology) | 62 | 15.5 |
| | Laboratory | 28 | 7 |
| | Radiology | 27 | 6.8 |
| | Pharmacy | 8 | 2 |
| **Education** | Associate's Degree | 12 | 3 |
| | Bachelor's | 253 | 63.3 |
| | Master's | 59 | 14.8 |
| | Ph.D. | 76 | 19 |

ease of use of the BYOD policy was the least influential factor for adopting the BYOD method from the staff's perspectives. (Fig 1)

Job performance improvement and accelerating tasks had the highest mean score in perceived usefulness dimension. The ease of remembering how to use the BYOD policy to access the HIS had the highest average in the perceived ease of use dimension. Safe access to patient medical information using the BYOD policy had the highest mean score in the perceived trust dimension. Training of employees regarding BYOD policy was determined as the most effective component in facilitating conditions. Information technology maintenance cost reduction in the BYOD policy had the highest mean score in the perceived cost-effectiveness dimension. (Table 2)

There was a positive and strong significant correlation between the perceived usefulness, perceived trust, perceived cost-effectiveness, facilitating conditions, and intention to adoption BYOD. Furthermore, there was a positive and moderate correlation between perceived ease of use and intention of adoption (P-value< 0.05). Pearson correlation test indicated that there was no significant correlation between age and work experience with the intention to adopt the BYOD policy (P-value> 0.05). In addition, the Spearman correlation test showed that there was no significant correlation between gender, education, and job and intention to adopt the BYOD method (P-value> 0.05). (Table 3)

## Discussion

Using personal mobile devices in the workplace is an effective and cost-effective approach to accessing healthcare services and information, facilitating healthcare professionals' access to knowledge and improving communication between caregivers [38–40]. Given the increasing penetration rate of active mobile phones in Iran [41], the establishment of the BYOD method and its effective management has the potential to improve the quality of healthcare and increase access to healthcare information in the country which often faces financial problems.

Perceived usefulness has been the most common factor in measuring mobile acceptance from healthcare professionals' perspective [42]. In this study, the mean factor of perceived usefulness of the BYOD was $3.76 \pm 1.01$ from the staff's point of view, which indicates that hospital staff believed that perceived usefulness had a significant influence on the adoption of this method. In addition, there was a positive and strong significant correlation between the perceived usefulness of BYOD and intention to adopt this policy (P-value> 0.01, r=0.802), which confirms the effect of perceived usefulness on BYOD acceptance. Moore [24] reported a mean of $5.07 \pm 1.72$ out of 7 for the perceived usefulness of BYOD, indicating a moderate influence of this factor on BYOD adoption. In addition, Wu et al. [43] in the study of mobile-based healthcare adoption from the healthcare staff's point of view obtained a mean of $4.89 \pm 1.15$ out of 5, indicating a high influence of this factor on acceptance of BYOD. The results of this study are in line with Wu et al. [43] and are relatively in line with Moore

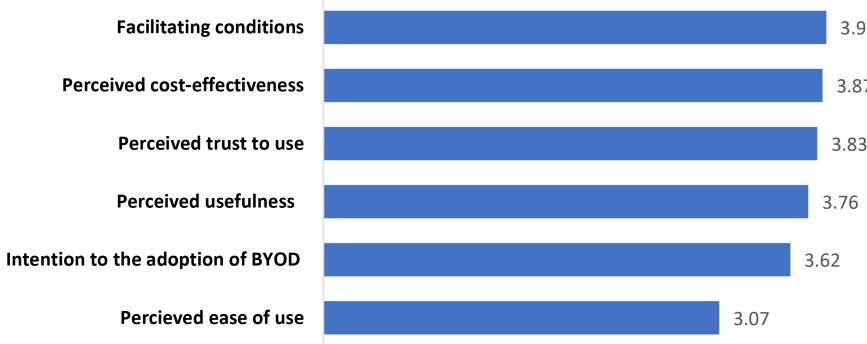

**Fig 1. The mean score of investigated factors from the hospital staff's perspective.**

**Table 2. Frequency distribution and mean score of the participants' answers to the items of the measured dimensions.**

| Variables | | Very Low | Below Average | Average | Above Average | Very High | Mean±S.D. |
|---|---|---|---|---|---|---|---|
| Perceived usefulness | Using the BYOD policy enables me to complete my tasks faster. | 20 (5) | 40 (10) | 57 (14.2) | 144 (36) | 139 (34.8) | **3.86±1.150** |
| | Using the BYOD policy supports critical aspects of my job. | 18 (4.5) | 56 (14) | 92 (23) | 153 (38.3) | 81 (20.3) | 3.56±1.098 |
| | Using the BYOD policy can increase my productivity. | 4 (1) | 47 (11.8) | 58 (14.5) | 203 (50.7) | 88 (22) | 3.81±.944 |
| | Using the BYOD policy can improve my job performance. | 4 (1) | 47 (11.8) | 62 (15.5) | 171 (42.8) | 116 (29) | **3.87±.995** |
| | Using the BYOD policy allows me to do more. | 24 (6) | 54 (13.5) | 46 (11.5) | 157 (39.3) | 119 (29.8) | 3.73±1.194 |
| | Using the BYOD policy can increase my effectiveness at work. | 6 (1.5) | 79 (19.8) | 24 (6) | 162 (40.5) | 129 (32.3) | 3.82±1.131 |
| | Using the BYOD policy has made things easier for me. | 2 (0.5) | 85 (21.3) | 21 (5.3) | 179 (44.8) | 113 (28.2) | 3.79±1.090 |
| | Using the BYOD policy has been useful in my work. | 8 (2) | 86 (21.5) | 50 (12.5) | 148 (37) | 108 (27) | 3.66±1.149 |
| Perceived ease of use | I find it difficult to use the BYOD policy. | 115 (28.7) | 136 (34) | 86 (21.5) | 53 (13.3) | 10 (2.5) | 2.27±1.090 |
| | Learning to work with the BYOD policy is easy for me. | 37 (9.3) | 41 (10.3) | 95 (23.8) | 156 (39) | 71 (17.8) | 3.46±1.169 |
| | Interacting with the hospital information system (HIS) using the BYOD policy is often challenging for me. | 74 (18.5) | 144 (36) | 95 (23.8) | 86 (21.5) | 1 (0.3) | 2.49±1.033 |
| | Using the BYOD policy to do whatever I need to do is simple. | 40 (10) | 51 (12.8) | 76 (19) | 167 (41.8) | 66 (16.5) | 3.42±1.197 |
| | The BYOD policy is inflexible and requiring a lot of mental effort for accessing healthcare information. | 65 (16.3) | 104 (26) | 129 (32.3) | 78 (19.5) | 24 (6) | 2.73±1.129 |
| | It is easy to remember how to use the BYOD policy to access the HIS. | 46 (11.5) | 41 (10.3) | 65 (16.3) | 162 (40.5) | 86 (21.5) | **3.50±1.257** |
| | My interaction with HIS using the BYOD policy is clear and understandable. | 50 (12.5) | 51 (12.8) | 62 (15.5) | 144 (36) | 93 (23.3) | 3.45±1.312 |
| | Acquiring enough skills to use the BYOD policy in the healthcare environment requires a lot of effort. | 93 (23.3) | 80 (20) | 65 (16.3) | 83 (20.8) | 79 (19.8) | 2.94±1.459 |
| | Acquiring enough skills to use the BYOD policy requires a lot of effort. | 68 (17) | 81 (20.3) | 76 (19) | 120 (30) | 55 (13.8) | 3.03±1.317 |
| | In general, the BYOD policy is easy for me to use. | 53 (13.3) | 30 (7.5) | 87 (21.8) | 137 (34.3) | 93 (23.3) | 3.47±1.290 |
| Perceived trust | It is easy to use the BYOD policy to access patient information. | 6 (1.5) | 24 (6) | 129 (32.3) | 119 (29.8) | 122 (30.5) | 3.82±.986 |
| | Using the BYOD policy to access medical information is safe. | 9 (2.3) | 34 (8.5) | 126 (31.5) | 100 (25) | 131 (32.8) | 3.78±1.066 |
| | I feel that healthcare organizations should allow the use of the BYOD policy to access medical information. | 35 (8.8) | 22 (5.5) | 65 (16.3) | 139 (34.8) | 139 (34.8) | 3.81±1.219 |
| | I feel that the BYOD policy is safe for accessing medical information. | 3 (0.8) | 30 (7.5) | 88 (22) | 159 (39.8) | 120 (30) | **3.91±.939** |
| Facilitating conditions | Training of employees will improve the adoption of the BYOD policy. | 2 (0.5) | 17 (4.3) | 86 (21.5) | 163 (40.8) | 132 (33) | **4.02±.873** |
| | Establishing clear guidelines consistent with hospital policies to use the BYOD policy will improve its adoption. | 4 (1) | 34 (8.5) | 95 (23.8) | 131 (32.8) | 136 (34) | 3.90±1.000 |
| | Having a specific mechanism for data storage will improve the BYOD policy acceptance. | 4 (1) | 17 (4.3) | 118 (29.5) | 148 (37) | 113 (28.2) | 3.87±.907 |
| | Providing the necessary devices and resources by the healthcare organization will improve the BYOD acceptance. | 8 (2) | 17 (4.3) | 99 (24.8) | 158 (39.5) | 118 (29.5) | 3.90±.941 |
| | Policy-making in the BYOD policy, information technology, and mobile device use will improve its adoption. | 4 (1) | 47 (11.8) | 79 (19.8) | 168 (42) | 102 (25.5) | 3.79±.986 |

*(Continued)*

**Table 2.** (Continued)

| Variables | | | Very Low | Below Average | Average | Above Average | Very High | Mean±S.D. |
|---|---|---|---|---|---|---|---|---|
| Perceived cost-effectiveness | | Using the BYOD policy is cost-effective. | 4 (1) | 74 (18.5) | 29 (7.2) | 166 (41.5) | 127 (31.8) | 3.85±1.095 |
| | | The cost of IT maintenance in the BYOD policy is lower than the IT costs of the traditional method. | 5 (1.3) | 14 (3.5) | 85 (21.3) | 181 (45.3) | 115 (28.7) | **3.97±.868** |
| | | In my opinion, the adoption of THE BYOD policy will have significant financial savings for the hospital. | 4 (1) | 72 (18) | 57 (14.2) | 138 (34.5) | 129 (32.3) | 3.79±1.111 |
| Intention to adoption | | I would like to recommend/suggest THE BYOD policy in the hospital. | 34 (8.5) | 58 (14.5) | 41 (10.3) | 147 (36.8) | 120 (30) | **3.65±1.277** |
| | | In my opinion, THE BYOD policy uses valid technology. | 5 (1.3) | 86 (21.5) | 113 (28.2) | 113 (28.2) | 110 (27.5) | 3.59±1.140 |

**Table 3.** The correlation between evaluated factors.

| Variables | | 1 | 2 | 3 | 4 | 5 | 6 | 7 | 8 | 9 |
|---|---|---|---|---|---|---|---|---|---|---|
| 1 | Perceived usefulness | 1 | | | | | | | | |
| 2 | Perceived ease of use | 0.524 | 1 | | | | | | | |
| 3 | Perceived trust to use | 0.818 | 0.526 | 1 | | | | | | |
| 4 | Perceived cost-effectiveness | 0.844 | 0.601 | 0.810 | 1 | | | | | |
| 5 | Facilitating conditions | 0.834 | 0.564 | 0.824 | 0.897 | 1 | | | | |
| 6 | Intention to the adoption of BYOD | 0.802 | 0.508 | 0.835 | 0.849 | 0.748 | 1 | | | |
| 7 | Age | −0.002 | −0.053 | −0.017 | −0.021 | 0.003 | −0.043 | 1 | | |
| 8 | Work experience | −0.004 | −0.071 | −0.015 | −0.035 | −0.008 | −0.061 | 0.923 | 1 | |
| 9 | education | −.020 | .012 | −.017 | .021 | −.009 | .009 | .129 | −.033 | 1 |

P- value<0.05 were considered statistically significant.

[24]. The reason for the very small difference between the present study and the Moore study may be due to the Likert scale used to measure factors influencing the intention to adoption the BYOD method.

The perceived ease of use has been the second most common factor influencing the acceptance of mobile devices in healthcare [42]. In the present study, the mean dimension of perceived ease of use was 3.07±0.48 from the healthcare staff's perspectives, which indicates the moderate effect of this dimension on the acceptance of the BYOD method. The obtained mean can indicate that it is easier for employees to use personal mobile phones in the healthcare environment to do organizational work because they are familiar with the personal device. Also, there was a moderate positive and significant correlation between the factor of perceived ease of use of the BYOD method and the intention to adopt this method (P-value>0.01, r=0.508). Wu et al. [43]revealed a mean of 4.88±1.13 for the perceived ease of use dimension. Moore et al. [24] showed a mean of 4.35±0.98 for this dimension. The present study is consistent with Wu et al. study, but it is contrary to Moore et al. study. Due to the increasing penetration rate of mobile phones in Iran and the greater skill of mobile phone users in the studied hospitals to work with their personal devices to perform organizational tasks, this discrepancy may have been caused. The relatively low mean score in this dimension may be influenced by challenges related to the usability of the BYOD policy, a lack of awareness among staff regarding its usefulness, and their resistance to adopting it. Improving staff training will enhance their comprehension of how the policy can benefit task performance and the quality of outcomes. Furthermore, providing organizational support for technical and security aspects of the devices in use can streamline the BYOD policy's ease of use.

The results of our study showed a mean of 3.83 ± 0.93 for perceived trust in using BYOD, which shows that employees strongly believe that the perceived trust affects the intention to adoption this policy. The positive, significant, and strong correlation between the perception of trust in the BYOD method and the intention to adoption this method by the staff also confirms these results (P-value< 0.05, r = 0.835). In Australia, Wani et al. [14] noted that the absence of BYOD policies in hospitals, along with concerns regarding security, privacy, and confidentiality, were the primary obstacles to adopting this strategy. Moqhbali et al. [44] found that there is a significant correlation between the perception of trust in the BYOD method and the intention to adopt it (P-value< 0.05, r = 0.247). Therefore, providing a secure infrastructure for staff's access to patients' health information will play a pivotal role in the intention to adoption of this method. Meng et al. [45] found that trust in mHealth services has a positive and important role in the intention to use this method. In his study, Moore [24] reported a mean of 5.19 ± 1.95 for the dimension of perceived trust in using the BYOD, which indicates a relative discrepancy with the present study. The existing discrepancy can be caused by the provision of a secure infrastructure to access and exchange data using the BYOD method in Moore's study.

Concerning facilitating conditions for the adoption of the BYOD method, a mean of 3.90 ± 0.87 was obtained, which indicates that the healthcare staff considered the facilitating conditions as the most important factor in the intention to adoption this method. The positive, strong, and significant correlation between facilitating conditions and intention to adopt the BYOD method also confirms this importance (P-value< 0.05, r = 0.748). Phichitchaisopa & Naenna [46] and Nanyombi and Habinka [47] emphasized that facilitating conditions play a decisive role in the adoption of health information technology. Nunes et al. [48] obtained an average of 5.07 ± 0.80 out of the overall average of 7 for the dimension of facilitating conditions for the adoption of mobile phones in healthcare, which is somewhat contrary to the results of the present study. This discrepancy can be seen in the smoothness of the facilitating conditions for using of BYOD policy in the study of Nunes et al. [48]. Therefore, training employees about the BYOD policy, developing guidelines by the organization's policies to use this method, specifying the mechanism of data storage and exchange, and providing part of the costs of the devices used by employees in the work environment can lead to the success of the use of the BYOD policy in healthcare organizations.

James and Griffiths [49] believe that the BYOD policy makes the use of this method cost-effective in the organization by making users more agile and increasing employee allegiance to the organization. In the present study, the mean of perceived cost-effectiveness of BYOD policy was determined to be 3.87 ± 0.97, which indicates that healthcare workers consider this method cost-effective. The positive, significant, and strong correlation between perceived cost-effectiveness and the intention to adoption of this policy (P-value<0.05, r = 0.849) also confirms the cost-effectiveness of using the BYOD method in healthcare organizations. Parinja et al. [50] showed that the use of mobile phones is a cost-effective way to improve the quality of healthcare services. Nanyombi and Habinka [47] reported that the costs of mobile devices reduce the acceptance of the BYOD method, because in the BYOD policy, all the costs of purchasing, maintaining, and updating the devices may be borne by the users. However, in general, previous studies indicate the cost-effectiveness of the BYOD method in healthcare organizations [51]. In this regard, providing maintenance costs and software updates for mobile devices used by the workplace by healthcare workers will facilitate the adoption of the BYOD method.

Understanding and intention to adoption of health information technology plays a pivotal role in the use of that technology. Sezgin et al. [52] expressed the intention of doctors to accept the BYOD method at about 60%. In this study, the average intention to adopt the BYOD method among the employees of the studied hospitals was determined as 3.62 ± 1.16. Moore reported a mean of 4.88 ± 1.66 from the overall average of 7 for intention to accept the BYOD method, which is relatively consistent with our results. Even though workers have a positive attitude towards adopting the BYOD policy, it is essential to take into account cybersecurity risks, device and network compatibility, along issues related to personal data protection and governance because of the sensitive nature of health information in hospital information systems. Wani et al. [14] emphasized that engaging stakeholders actively, developing security protocols suited to the requirements of clinical personnel, ensuring BYOD policies are in harmony with workflow needs, fostering proactive collaboration

between technical and clinical teams, educating clinical staff, and cultivating a robust cybersecurity culture are essential steps for enhancing security within the BYOD framework.

## Study Strengths and limitations

In this research, a thorough and precise statistical analysis was conducted, adhering to robust research methods, to gain scientific understanding of the factors that influence employees' intention to adopting the BYOD policy in teaching hospitals. Additionally, the study has offered solutions to address the current challenges. As a result, decision-makers and administrators can leverage the insights gained to formulate the required policies. However, our research has several constraints. Firstly, the study was conducted with healthcare professionals from five teaching hospitals within a single province, which may limit the generalizability of the findings to all hospitals across Iran. It would be beneficial to carry out a similar study with a larger participant pool from various provinces to enhance the cluster sampling methods' generalizability. Secondly, despite much effort and motivation by the research team, the number of physicians involved in this study was limited. As a result, the findings cannot be extrapolated to all physicians within the hospitals examined; this might improve by employing a larger sample size for distinct population subgroups in future studies. Thirdly, this research relies exclusively on quantitative data, which hinders a thorough understanding of the perceptions of healthcare workers. Therefore, it is advisable to undertake qualitative research to more accurately identify the motivations or underlying issues related to the implementation of the BYOD policy in Iranian hospitals. Fourthly, our study lacks a comprehensive examination of information security and related dimensions, including privacy and confidentiality of patient information, in using BAYOD. Given the sensitive nature of healthcare industry data, future studies must focus on these dimensions and identify solutions to improve information security using BAYOD in healthcare institutions.

## Conclusion

Perceived cost-effectiveness, facilitating conditions, perceived trust, perceived usefulness, and perceived ease of use have been determined as the most influencing factors in the intention to adoption of the BYOD method from the healthcare workers' perspective. Although workers in the examined hospitals generally have a favorable view about intention to adoption of BYOD policy, it is crucial to consider cybersecurity threats, device and network interoperability, as well as matters concerning personal data protection and governance due to the sensitive nature of health information in these facilities. Ensuring the security of access to healthcare information, provision, support and maintenance of devices used by employees in the workplace for work-related activities can play a significant role in promoting the intention to adopt the BYOD. Our results can be useful for planning and policy-making to increase the adoption and acceptance of the BYOD method in healthcare institutions.

## Acknowledgments

The authors thank the healthcare workers who participated in this study for sharing their valuable experiences.

## Author contributions

**Conceptualization:** Jahanpour Alipour, Abolfazl Payandeh, Afsaneh Karimi.

**Data curation:** Jahanpour Alipour, Abolfazl Payandeh, Afsaneh Karimi.

**Formal analysis:** Jahanpour Alipour, Abolfazl Payandeh, Afsaneh Karimi.

**Methodology:** Jahanpour Alipour, Abolfazl Payandeh, Afsaneh Karimi.

**Writing – original draft:** Jahanpour Alipour, Abolfazl Payandeh, Afsaneh Karimi.

**Writing – review & editing:** Jahanpour Alipour, Abolfazl Payandeh, Afsaneh Karimi.

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
