## [Decision Letter · Decision Letter 0]

PONE-D-24-55967Factors influencing the adoption of the BYOD policy in teaching hospitals: A cross-sectional study from Southeastern IranPLOS ONE

Dear Dr. Alipour,

Thank you for submitting your manuscript to PLOS ONE. After careful consideration, we feel that it has merit but does not fully meet PLOS ONE’s publication criteria as it currently stands. Therefore, we invite you to submit a revised version of the manuscript that addresses the points raised during the review process.

We look forward to receiving your revised manuscript.

Kind regards,

Mariam Rehman, Ph.D

Academic Editor

PLOS ONE

Journal Requirements:

2. Please remove all personal information, ensure that the data shared are in accordance with participant consent, and re-upload a fully anonymized data set.

3.  We are unable to open your Supporting Information file BYOD_for Analysis.sav. Please kindly revise as necessary and re-upload.

Reviewers' comments:

Reviewer's Responses to Questions

**Comments to the Author**

1. Is the manuscript technically sound, and do the data support the conclusions?

Reviewer #1: Yes

Reviewer #2: Yes

2. Has the statistical analysis been performed appropriately and rigorously? 

Reviewer #1: Yes

Reviewer #2: Yes

3. Have the authors made all data underlying the findings in their manuscript fully available?

Reviewer #1: Yes

Reviewer #2: Yes

4. Is the manuscript presented in an intelligible fashion and written in standard English?

Reviewer #1: Yes

Reviewer #2: Yes

5. Review Comments to the Author

Reviewer #1: The topic of the article was intriguing from my perspective. BYOD in hospitals can lead to cost reduction and increased productivity. However, the adoption of this technology faces numerous challenges, such as security concerns and acceptance by staff. Given the significance of this topic in the field of health information technology, the article addresses these challenges clearly and accurately. Additionally, the appropriate sampling and the large sample size (1,130 individuals from five hospitals) enhance the credibility of the results, making it a noteworthy article.

Overall:

The methodology section should specify how hospitals were selected. While the article provides a general overview of the research methodology, it does not address potential limitations or sampling biases (e.g., selection of specific hospitals or types of staff).

One of the challenges of BYOD in hospitals is data security. The article does not explore security issues. Given the sensitivity of medical information and health data, this aspect appears to be essential to consider.

The study primarily utilized quantitative data; I believe that qualitative analyses, such as interviews, might have been more effective in understanding staff attitudes and experiences.

In the conclusion and recommendations section, I think the article could be improved by offering more practical suggestions for hospitals and IT officials. It could present recommendations for better implementation of BYOD in hospitals.

Some more recent and relevant sources from the field of health information technology and BYOD acceptance could be added to demonstrate the research's timeliness.

Reviewer #2: 1. Although the study offers valuable insights, the sample is drawn from hospitals within a single province, and the representation of certain groups (e.g., physicians) is limited. The authors should discuss more explicitly how these sampling limitations might affect the generalizability of the findings and propose strategies for future research (e.g., expanding the study to multiple provinces or increasing subgroup representation).

2. While the manuscript uses TAM constructs effectively, the discussion could benefit from a deeper integration of recent literature specifically addressing BYOD in healthcare. Strengthening the theoretical framework with current studies could help position the work within broader debates about technology adoption in healthcare. The authors can consider other TAM-related studies in other fields such as "Moderating effects of policy measures on intention to adopt autonomous vehicles: Evidence from China" and "Analyzing factors influencing IoT adoption in higher educational institutions in Saudi Arabia using a modified TAM model".

3. The relatively low mean score for perceived ease of use merits further discussion. The authors might explore potential reasons for this finding (such as usability challenges or resistance to change among staff) and consider how this aspect could be addressed in practice.

4. More detail on the process of questionnaire development and validation (including potential biases or limitations in measurement) would be beneficial. For example, clarifying how the CVR and CVI thresholds were determined and discussing any steps taken to mitigate measurement error would strengthen the methodology section.

6. PLOS authors have the option to publish the peer review history of their article (what does this mean? ). If published, this will include your full peer review and any attached files.

**Do you want your identity to be public for this peer review?** For information about this choice, including consent withdrawal, please see our Privacy Policy .

Reviewer #1: No

Reviewer #2: No

---

## [Author Response · Author response to Decision Letter 1]

8 Mar 2025

Dear Reviewers,

Thank you very much for your consideration. We really appreciate the comments and have learned a lot. According to the suggestions of the reviewers and editor, appropriate changes were made in the revised manuscript.

Responses reviewers’ comments:

Reviewer #1: The topic of the article was intriguing from my perspective. BYOD in hospitals can lead to cost reduction and increased productivity. However, the adoption of this technology faces numerous challenges, such as security concerns and acceptance by staff. Given the significance of this topic in the field of health information technology, the article addresses these challenges clearly and accurately. Additionally, the appropriate sampling and the large sample size (1,130 individuals from five hospitals) enhance the credibility of the results, making it a noteworthy article.

Overall:

1. The methodology section should specify how hospitals were selected. While the article provides a general overview of the research methodology, it does not address potential limitations or sampling biases (e.g., selection of specific hospitals or types of staff).

Response: We appreciate the reviewer's valuable scientific comments and constructive suggestions, which help us to improve the quality of the manuscript.

Regarding how the hospitals were selected, we selected all teaching hospitals affiliated with Zahedan University of Medical Sciences. We mentioned this in the method section for clarity. Page 4, line 107.

Regarding sampling limitations and biases, we reduced the possible measurement error by using an approximately large sample, careful design of the sampling, data collection, and analysis procedures. Extra information was added accordingly. Pages 4 and 5, lines 119-123.

One of the challenges of BYOD in hospitals is data security. The article does not explore security issues. Given the sensitivity of medical information and health data, this aspect appears to be essential to consider.

Response: We agree with the reviewer and appreciate the reviewer’s scientific comment and suggestion.

Given the sensitivity of healthcare data, information security and its related dimensions, including privacy and confidentiality of patient information, are very important and sensitive issues. In the present study, we briefly addressed this issue in the dimension of perceived trust to some extent. We also mentioned it in the discussion section. However, a comprehensive examination of the all dimensions of security, privacy, and confidentiality of patient information in the implementation and use of the BAYOD policy in healthcare institutions seems more than necessary, and future studies focusing on this area are needed. We have added this as a limitation of the present study in the study limitations section. Page 13, lines 304-308.

The study primarily utilized quantitative data; I believe that qualitative analyses, such as interviews, might have been more effective in understanding staff attitudes and experiences.

In the conclusion and recommendations section, I think the article could be improved by offering more practical suggestions for hospitals and IT officials. It could present recommendations for better implementation of BYOD in hospitals.

Response: We agree with the reviewer, we had mentioned this in the study limitations section and suggested that future studies focus on qualitative methods. A qualitative study of this topic is also part of the authors' future research plans. Page 13, lines 301-304.

Some more recent and relevant sources from the field of health information technology and BYOD acceptance could be added to demonstrate the research's timeliness.

Response: We agree with the reviewer and we used five new relevant references that published in 2024 or 2025.

Reviewer #2: 1. Although the study offers valuable insights, the sample is drawn from hospitals within a single province, and the representation of certain groups (e.g., physicians) is limited. The authors should discuss more explicitly how these sampling limitations might affect the generalizability of the findings and propose strategies for future research (e.g., expanding the study to multiple provinces or increasing subgroup representation).

Response: Revised and extra information added. Page 13, Lines 294-304.

2. While the manuscript uses TAM constructs effectively, the discussion could benefit from a deeper integration of recent literature specifically addressing BYOD in healthcare. Strengthening the theoretical framework with current studies could help position the work within broader debates about technology adoption in healthcare. The authors can consider other TAM-related studies in other fields such as "Moderating effects of policy measures on intention to adopt autonomous vehicles: Evidence from China" and "Analyzing factors influencing IoT adoption in higher educational institutions in Saudi Arabia using a modified TAM model".

Response: We appreciate the reviewer for the constructive comment, we used some new references in the discussion section and we cited the studies mentioned by the reviewer in the theoretical framework section of the manuscript to strengthen the theoretical framework of our manuscript. Page 4, line 99; page 11, lines 235-237, page 12, lines 282-87.

3. The relatively low mean score for perceived ease of use merits further discussion. The authors might explore potential reasons for this finding (such as usability challenges or resistance to change among staff) and consider how this aspect could be addressed in practice.

Response: We agree with the reviewer’s comment. We added explanations in the discussion section about the reason for the low score of the referenced dimension and how to improve the situation. Page 11, lines 255-230.

4. More detail on the process of questionnaire development and validation (including potential biases or limitations in measurement) would be beneficial. For example, clarifying how the CVR and CVI thresholds were determined and discussing any steps taken to mitigate measurement error would strengthen the methodology section.

Response: We agree with the reviewer. we add extra information accordingly.

Regarding sampling limitations and biases, we reduced the possible measurement error by using an approximately large sample, careful design of the sampling, data collection, and analysis procedures. Pages 4 and 5, lines 119-123.

A reference was added for the CVI and CVR thresholds clarity. Page 5, line 141.

---

## [Decision Letter · Decision Letter 1]

Factors influencing the adoption of the BYOD policy in teaching hospitals: A cross-sectional study from Southeastern Iran

PONE-D-24-55967R1

Dear Dr. Alipour,

We’re pleased to inform you that your manuscript has been judged scientifically suitable for publication and will be formally accepted for publication once it meets all outstanding technical requirements.

Kind regards,

Asli Suner Karakulah, PhD

Academic Editor

PLOS ONE

Additional Editor Comments (optional):

Reviewers' comments:

Reviewer's Responses to Questions

**Comments to the Author**

1. If the authors have adequately addressed your comments raised in a previous round of review and you feel that this manuscript is now acceptable for publication, you may indicate that here to bypass the “Comments to the Author” section, enter your conflict of interest statement in the “Confidential to Editor” section, and submit your "Accept" recommendation.

Reviewer #2: All comments have been addressed

2. Is the manuscript technically sound, and do the data support the conclusions?

Reviewer #2: Yes

3. Has the statistical analysis been performed appropriately and rigorously? 

Reviewer #2: Yes

4. Have the authors made all data underlying the findings in their manuscript fully available?

Reviewer #2: Yes

5. Is the manuscript presented in an intelligible fashion and written in standard English?

Reviewer #2: Yes

6. Review Comments to the Author

Reviewer #2: The authors have addressed all my concerns. I have nothing to add to this comment to authors. No further questions.

7. PLOS authors have the option to publish the peer review history of their article (what does this mean? ). If published, this will include your full peer review and any attached files.

**Do you want your identity to be public for this peer review?** For information about this choice, including consent withdrawal, please see our Privacy Policy .

Reviewer #2: No
